# Designing *Aedes* (Diptera: Culicidae) Mosquito Traps: The Evolution of the Male *Aedes* Sound Trap by Iterative Evaluation

**DOI:** 10.3390/insects12050388

**Published:** 2021-04-27

**Authors:** Kyran M. Staunton, Jianyi Liu, Michael Townsend, Mark Desnoyer, Paul Howell, Jacob E. Crawford, Wei Xiang, Nigel Snoad, Thomas R. Burkot, Scott A. Ritchie

**Affiliations:** 1College of Public Health, Medical and Veterinary Sciences, James Cook University, Smithfield, QLD 4878, Australia; michael.townsend@jcu.edu.au (M.T.); tom.burkot@jcu.edu.au (T.R.B.); scott.ritchie@jcu.edu.au (S.A.R.); 2Australian Institute of Tropical Health and Medicine, James Cook University, Smithfield, QLD 4878, Australia; 3Verily Life Sciences, Debug, 259 East Grand Avenue, South San Francisco, CA 94080, USA; jianyil@verily.com (J.L.); mdesnoyer@verily.com (M.D.); paulihowell@verily.com (P.H.); jacobcrawford@verily.com (J.E.C.); nigelsnoad@verily.com (N.S.); 4School of Engineering and Mathematical Sciences, La Trobe University, Melbourne, VIC 3083, Australia; W.Xiang@latrobe.edu.au

**Keywords:** *Aedes aegypti*, mosquito trap, sound trap, male, dengue, sound lure

## Abstract

**Simple Summary:**

This study presents a product profile and describes key developmental trial results concerning the invention of the male *Aedes* sound trap. This trap contains a sound lure which mimics female mosquito wingbeat frequencies to attract male *Aedes aegypti*. Male *Ae. aegypti* capture rates were significantly influenced by the presence of, and large variations in size of, a black trap base. Male capture rates were also influenced by reducing the trap entrance below 2.5 cm (1 inch), but not by variations in sound lure volume, between 63–78 dB, or whether the sound lure tone was played continuously or intermittently. No differences were noted in catch rates of male *Ae. aegypti* in traps using sound lures set to 450 Hz or 500 Hz, but younger adult males were captured at lower rates than some older male groups. Lastly, when the trap was exposed to windy conditions male *Ae. aegypti* capture rates were affected to varying degrees, depending on trap orientation relative to the wind and whether the wind was continuous or intermittent. The trap profile and behavioural findings associated with this trap development are relevant to ensuring effective surveillance using this new tool, as well as the development of other mosquito traps.

**Abstract:**

Effective surveillance of *Aedes aegypti* (Linnaeus, Diptera: Culicidae) is critical to monitoring the impact of vector control measures when mitigating disease transmission by this species. There are benefits to deploying male-specific traps, particularly when a high level of catch-specificity is desired. Here, the rationale behind the developmental process of an entirely new trap which uses a sound lure to capture male *Ae. aegypti*, the male *Aedes* sound trap (MAST), is presented as a target product profile with findings from developmental trials of key trap components and performance. Trial results suggest that the presence of a black base associated with the trap influenced male catches as did variations in size of this base, to a degree. Trap entrance shape didn’t influence catch rates, but entrance size did. No significant differences in catch rates were found when sound lures were set to intermittent or continuous playbacks, at volumes between 63–74 dB or frequencies of 450 Hz compared to 500 Hz. Additionally, adult males aged 3 days post-eclosion, were less responsive to sound lures set to 500 Hz than those 4 or 6 days old. Lastly, almost no males were caught when the MAST directly faced continual winds of 1.5 ms^−1^, but males were captured at low rates during intermittent winds, or if the trap faced away from the wind. The developmental process to optimising this trap is applicable to the development of alternate mosquito traps beyond *Aedes* sound traps and provides useful information towards the improved surveillance of these disease vectors.

## 1. Introduction

*Aedes aegypti* is a primary vector of dengue, Zika, and yellow fever viruses and are expanding in global distribution [1,2,3]. With this escalating burden to public health, improved *Aedes* control is of increasing importance to public health staff, and surveillance of vectors is critical to effective control [4]. There are many mosquito traps available targeting a range of different species at different life stages and environmental conditions. While some traps are more broadly used than others, each product has its own benefits and limitations and one rarely suits all programmatic needs. Consequently, despite mosquito trap development being an extensive field of research for decades [5], there is still a need for continued investment in the creation of new traps.

Mosquito traps are generally designed to specifically target either egg or adult life stages. Ovitraps (egg-traps) are deployed with water combined with an olfactory cue to attract gravid females which lay their eggs on a specialised substrate incorporated in the trap [6]. While these traps are generally very simple and easy to deploy, they require frequent servicing and the eggs must be hatched in a laboratory so that larvae or adults can be reared and examined for species to be determined. Unfortunately, the laboratory rearing and larval identification required with ovitraps takes time and additional resources. Traps targeting adult mosquitoes may be more expensive than ovitraps, but they retain samples which, if in good condition, can be immediately identified to species.

The current suite of adult mosquito traps deployed for surveillance activities are designed to capture female mosquitoes, which bite and therefore transmit diseases. Such traps exploit female mosquito behaviours such as harbouring, host-seeking or ovipositing. Traps targeting harbouring *Ae. aegypti* leverage the high sensitivity of this species to light [7] and their attraction to dark objects of low reflectance [8,9,10]; therefore, such traps often use dark colours to attract this species [11]. Traps built for host-seeking females may utilise chemical lures such as CO_2_ or human skin scent mimics such as the BG-Lure [12,13], which mimic chemicals that would normally be excreted by hosts. Such lures can significantly improve mosquito catch rates [12], but will increase surveillance costs and can be operationally demanding to deploy. Lastly, traps may also target gravid females, such as the BioGents-Gravid Aedes Trap (BG-GAT; Biogents, Regensburg, Germany) [14] and the Centres for Diseases Control and Prevention (CDC) autocidal gravid ovitraps (AGO) [15], which are set with water that is often infused with grass or hay to mimic attractive larval habitats where females seeking on oviposition site may enter to lay their eggs.

The BG-GAT and AGO both rely on passive trapping systems where the mosquitoes enter the capture chambers of their own volition. These passive systems can be built using cheap, simple and highly durable components and don’t require electricity to operate. On the other hand, an active trapping system, such as operating a fan to draw in mosquitoes at the trap entrance (e.g., in the BG-Sentinel (Biogents, Regensburg, Germany) [12] or Centres for Diseases Control and Prevention (CDC) miniature light trap [16]), is a common alternative to a passive entrance system. Fans are extremely effective insect capture tools, however they require continuous power supplies and indiscriminately catch other invertebrates, increasing operating costs and extending laboratory resources required to identify catches [17].

Securing the mosquitoes once they have entered the trap is an important consideration for all mosquito trap development. Specimens can be held using the suction force of a fan [12], by being knocked down with insecticide [14] or by being physically stuck due to glue or oil [18]. There are costs and benefits to all approaches. Fans are extremely effective at securing catches, but can desiccate and destroy samples over time [19]. Insecticide can be fast acting and long lasting, but with the development of insecticide-resistance in mosquitoes in many regions of the world, these chemicals are often ineffective [18]. Lastly, fixing substances such as oil or glue may be effective, but are often frustrating to service [19].

While female mosquitoes are of primary interest to public health staff, male mosquitoes still represent a significant proportion of any population [20] and can be sampled to indicate species presence. Additionally, male *Ae. aegypti* and *Aedes albopictus* are being mass reared and released in control programs therefore organisations are concerned with monitoring their densities as efficiently as possible [21,22]. There are also a range of design features regarding male mosquito traps which may be of interest to certain mosquito programs. For example, unlike females, male *Ae. aegypti* will selectively respond to sound lures tuned to specific female mosquito’s wingbeat frequencies [23,24] and can therefore be attracted to enter passive trap entrances [25]. This means that substantial power savings can occur with these designs as without the power requirements of fans [26,27] sound traps can be operated with only one or two AA batteries [17,27]. Additionally, male mosquito traps can be designed to be highly species-specific therefore reducing sorting and identification requirements [17].

We developed an entirely new sound trap, the male *Aedes* sound trap (MAST), to capture male *Ae. aegypti* in a manner considered more fit for purpose than previous male mosquito traps [17]. This trap was field-tested locally in both Cairns and the Torres Strait, Australia [17,28], patented [29] and further tested in Papua New Guinea, Mexico, and Belize [30]. Trap characteristics relevant to the development of the MAST are captured in the Target Product Profile (TPP; Table 1). These considerations were outlined to guide developmental trials and optimise trap design and performance. Here we report on key laboratory trials performed in relation to the TPP to demonstrate the iterative process required to create an *Aedes* trap prototype.

## 2. Methods

### 2.1. Experimental Conditions and Standard Trial Methods

MAST development experiments (Appendix A) were performed in one of three different settings in Cairns, Queensland Australia. Small scale trials were performed within a square based domed tent (1.8 m sides, 2 m height; Wild Country, Tideswell, England) inside a controlled temperature room (Ritchie, et al. [31]), maintained at 28 °C and 70% RH. For each trial, 20 virgin males (separated by sex as pupae), generally 4–6 days post-eclosion, were released and capture rates determined after 30 min. The trap was taken to the lab bench and any mosquitoes still flying inside the head were knocked down with CO_2_. All uncaught males were removed from the tent after each trial. Treatments were always randomly rotated within the tent throughout the trial period.

The second setting was larger semi-field trials in cages measuring 13.7 m × 5.5 m × 4 m high—described in detail as ‘Cage A’ by Darbro, et al. [32]. These trials included ca. 2000 unseparated adult male and female *Ae. aegypti* with expected ratios of 50% male: female (4–6 days post emergence), per experiment, that were released at least one hour prior to the trial into the semi-field cage between 08:30 and 10:30 a.m., when temperatures were >26 °C. Traps to be compared were placed randomly in each corner of the flight cage with their entrances facing inwards (not directly towards a wall) for a five-minute trial (catch rates were so high that longer periods would trap out the population too quickly). A maximum of four treatments were compared at once. At the end of each trial, traps were removed from the flight cages and any mosquitoes still flying inside the trap head were knocked down with CO_2_. All mosquitoes captured were sorted to sex and counted. The traps were returned to the flight cages in randomly allocated new positions and this process repeated until 3 Latin squares were completed. As with tent trials, temperature and humidity were recorded with wireless sensor tags (Cao Gadgets LLC, Irvine, CA, USA).

The short trial durations associated with both tent and semi-field cage trials were not considered to comprehensively replicate field conditions where, for example, males may potentially take much longer to be successfully caught in the traps or be distracted by competing cues. They were however considered to still provide useful information as to the relative attractiveness of each trap modification.

The third setting were field trials in Cairns urban landscape either at a premises containing a large traditional Queenslander styled house located within the Cairns’ suburb of Edge Hill or as part of Latin Squares described by Staunton, et al. [17].

Mosquitoes tested in both tent and semi-field cage trials were selected from the F1–F3 generations of *Ae. aegypti* reared from eggs collected in Cairns using ovitraps. These mosquitoes were infected with the *w*Mel strain of *Wolbachia* due to previous releases throughout this area by the World Mosquito Program [33]. These mosquitoes were maintained in populations of ~300 in 30 cm × 30 cm cages in a controlled temperature room at 28 °C and 70% RH, separate to the room with the tent. Larvae were fed on ground fish food (TetraMin Tropical Flakes, Melle, Germany) at a cohort density of ~500/3 L bucket. Twenty male pupae or 100 pupae of mixed sexes were put into cups (11 cm wide × 10 cm deep with 1 cm of water in the bottom) for tent or semi-field trials, respectively. All adults had access to 50% honey and water solutions. Human ethics approval was granted by the James Cook University ethics committee, H6286.

### 2.2. Standard Trap Design

MASTs consist of a clear plastic capture container (head), with a sound lure inside and a black base on which the MAST head sits. The first prototype MAST design (MAST 1), which was generally used in these trials, consisted of a translucent 4 L rectangular plastic head (18.7 cm (H) × 18.7 cm (W) × 17.2 cm (D); Princeware, India) with 2 × 5 cm^2^ entrances cut from the centre base of neighbouring sides (Figure 1). The sound lure inside the head utilised an 8 ohm, 0.5 watt speaker which produced a sinusoidal tone and was mounted to a printed circuit board (PCB). The PCB included a light detector and physical buttons enabling the sound produced to be adjusted by volume, frequency and whether or not it played continuously, at 30 s intervals and/or only during the day [17]. The lure was powered by 2 alkaline AA batteries and the front of the speaker was protected by a plastic grill. Unless otherwise specified, the sound lure was set to a tone of 500 Hz at 63 dB at trap entrance and played this tone intermittently for 30 s on-off intervals (Appendix A). All sound lure speakers were covered by BG-GAT sticky cards (Biogents, Regensburg, Germany) cut to 8 × 7 cm, fixed to the plastic grill using Blu Tack (Bostick, Milwaukee, WI, USA). The MAST is designed to house a sensor system whereby captures are recorded and communicated to public health staff. An early concept of this sensor system utilised a phone; therefore, a ‘mock phone’ consisting of a piece of wood, cut to the size of a smart phone (14.5 cm (H) × 7 cm (W) × 0.9 cm (D)) and wrapped in white tape, was placed inside the capture container. The head was placed on a base constructed from two crossed black corflute (corrugated polypropylene 5 mm thickness; Corex Plastics Australia; Figure 1). The corflute sections were cut out as 40 cm squares then, at 20 cm high, a triangular section was cut out so that they were tapered inwards to create a top side of 20 cm in length. A second MAST prototype (MAST 2) consisted of a 2.5 L rectangular clear plastic head on 2 black stacked grower’s pots [17].

### 2.3. Experiment 1. MAST Base Height—Semi-Field Cage Trial

MAST 1 versions were deployed, each with a sound lure set in the centre of the container facing halfway between the entrances on 18 August 2018. Catch rates of male *Ae. aegypti* in four trap types were compared, one without a base and others with corflute bases which were 40 cm, 60 cm, and 80 cm high (Appendix A).

### 2.4. Experiment 2. MAST Base Height—Single Premises Field Trial

Between 12–19 January 2019, daily trials were performed at a single premise in the suburb of Edge Hill, Cairns, Queensland Australia. In this trial, MAST 2 versions (described by Staunton, et al. [17]), with different sized bases (short = 23 cm high (BG-GAT base) and tall = 45 cm high (standard stacked MAST 2 bases)), were rotated on opposite sides of a house (15 m apart) to catch male *Ae. aegypti*. The traps were set for two hours in the morning and then rotated for an additional two hour period in the afternoon. This method was repeated for 6 days until each trap had been exposed to either location during morning or afternoon 6 times (*n* = 12).

### 2.5. Experiment 3. MAST Base Height—Multiple Premisses Field Trial

Due to the conflicting observations between the cage and single premises trials, we decided to run an additional field experiment where male *Ae. aegypti* were exposed to MASTs for longer periods of time (one week) and in multiple (12) locations. Between 14 February and 14 March 2019 we compared MASTs with tall and short bases in a Latin square design. The experimental design and data regarding the tall MASTs has been previously described as ‘MAST Spray’ in ‘Trial Two’ by Staunton, et al. [17]. The version of MAST head used on a short base was identical to MAST Spray described by Staunton, et al. [17], but instead of having a base which measured 45 cm high it was only set the larger of the two buckets used which was 27 cm high.

### 2.6. Experiment 4. MAST Head—Various Entrance Numbers, Sizes, and Shapes Semi-Field Cage Trial

All mosquito colonies were maintained as described above, although F7 adults were used and test subjects were reared and released from the cups into the small flight cage A as described above. On 22 August 2018, we tested male *Ae. aegypti* catch rates in MASTs with four different entry options cut out of the centre bases of the MAST 1 heads: (1) two squares, each with 5 cm sides, (2) two squares, each with 2.5 cm sides, (3) one square with 2.5 cm sides and (4) one upside down equilateral triangle with 2.5 cm sides (Appendix A). For MAST heads with single entrances, sound lures faced directly towards entrances, whereas for heads with two entrances sound lures faced in the middle between both openings.

### 2.7. Experiment 5. MAST Head—Various Entrance Sizes Semi-Field Cage Trial

Mosquitoes tested derived from F3 generation maintained in colonies and released into the flight cage on 6 September 2018 at four days post emergence, as described above. MAST 1 versions were deployed with four differently sized inverted equilateral triangular entrances cut out of the middle centre of one size with the sound lure placed directly behind. The triangular entrances tested had sides of either 1 cm, 1.5 cm, 2 cm, or 2.5 cm (Appendix A). Sound volume was recorded at the trap entrance using an SD-4023 sound level meter (Reed Instruments, Los Angeles, CA, USA).

### 2.8. Experiment 6. Sound Lure—Continuous vs. Intermittent Tones Semi-Field Cage Trial

Mosquitoes tested derived from F3 generation maintained in colonies and released into the flight cage on 26 September 2018 at four days post emergence, as described above. MAST 1 versions were used with only a single inverted triangular entrance with 2.5 cm sides and a sound lure placed directly behind the entrance. The sound lures either produced tones continually or intermittently, with a 30 s on/off pattern. Two duplicates of each trap type were randomly rotate through two 4 × 4 Latin Squares.

### 2.9. Experiment 7. Sound Lure—Various Volumes Semi-Field Cage Trial

All mosquito colonies were maintained as described above with F3 adults used. Three MAST 1 traps were set in the flight cage, on 30 October 2018, with sound lures playing intermittent tones (30 s on/off) at 500 Hz. Traps were set to either the lowest (63 dB), middle (68 dB) and highest (74 dB) volume settings, determined using an SD-4023 sound level meter (Reed Instruments, California) held directly at the trap entrance.

### 2.10. Experiment 8. Sound Lure—Various Frequencies Tent Trial

All mosquito colonies were maintained as described above with F3 adults being used. Test subjects were reared and kept in cups as described above until being released into the tent in CT 1 for each trial between 15–18 October 2018. The traps we used in these trials comprised of the MAST 1 version of head placed on the 45 cm grower’s pots base used with the MAST 2 (Appendix A). We predominantly tested for any difference in catch rates between 450 and 500 Hz (*n* = 23), but also ran four indicative trials to note the catch rates at 550 Hz, 600 Hz and with the speaker turned off. We also tested for any differences in catch rates of males at 3, 4, 5 and 6 days post-eclosion in the MAST with sound lures set to either 450 or 500 Hz (*n* = 10–12, v1 lure, vol 63 dB, tone intermittent).

### 2.11. Experiment 9. Influence of Wind on Catch Rates Tent Trial

All mosquito colonies were maintained as described with F3 adults, aged from 3–6 days old post-eclosion, being used. Test subjects were reared and kept in cups as described above until being released into the tent in CT 1 for each trial between 28–31 August 2018. MAST 1 was utilised with a corflute base and the sound lure set to 500 Hz, 63 dB and playing an intermittent tone (30 s intervals). Wind was produced using an oscillating fan (Target *Essentials* 40 cm pedestal fan, China; Appendix A) and recorded with a Kestrel 1000 anemometer (Nielsen–Kellerman, Boothwyn, PA, USA). We ran trials investigating the influence on male *Ae. aegypti* catch rates in the MAST exposed to wind conditions of: wind blowing continuously on the trap entrance (wind speed at entrance ~1.5 m/s), wind blowing continuously on the back of the MAST head (wind speed at entrance 0 m/s) and wind blowing intermittently on the trap entrance (wind speed at entrance 1.5 m/s at 9 s intervals). As a control, we also tested male *Ae. aegypti* catch rates in the MAST with the fan turned off.

### 2.12. Data Analysis

All statistical investigations were performed in the R statistical environment ver 3.5.3 [34]. For tent trials, parameters (‘trap type’ and ‘mosquito age’ for experiment 8 and ‘wind treatment’ for experiment 9) were fitted to a response variable of the proportion of males caught using a Generalized Linear Model (GLM) with a binomial distribution (experiment 8) and a quasibinomial distribution when the binomial distribution was overdispersed (experiment 9) and a logit link function with the *stats* package [34]. For experiment 8, an interaction term between parameters was initially modelled, but found to be not significant so was removed in the final model presented. The effect of predictors were analysed for significance using analysis of deviance in the *car* package [35]. If significant differences were detected then the marginal means of groups were compared using Tukey post-hoc analyses with the *emmeans* package [36].

For trials performed in the flight cages, parameters ‘trap type’ and ‘location’ were fit to the response variable of the number of male *Ae. aegypti* captured in each trap by a GLM with a Poisson distribution (experiments 4, 5, 6 and 7) and a log link function using the *stats* package [34] and, when overdispersed (experiment 1), a negative binomial distribution and a log link function using the *MASS* package [37]. For experiment 5 the parameter ‘sound volume’ was also fit to the model. Additionally, an offset of the total number of males caught in all traps run per trial to account for variation in catch rates during the day (primarily as males were not replaced after each trial, but also because environmental conditions often changed throughout the day and may have affected catch rates per trial). Further analyses determining the difference between groups by analysis of deviance and Tukey post-hoc tests were performed as stated above.

For field trials, again parameters (‘trap type’, ‘location’ and either ‘time of day’ for experiment 2 or ‘week’ for experiment 3) were fitted to the response variable of the number of male *Ae. aegypti* captured in each trap per trial by a GLM with a negative binomial distribution (both data sets were overdispersed with Poisson distributions) and a log link function using the *MASS* package [37]. As above, when suitable, further analyses determining the difference between groups by analysis of deviance and Tukey post-hoc tests were performed.

## 3. Results

### 3.1. Experiment 1. MAST Base Height—Flight Cage

A total of 548 males and 9 females were caught in this experiment with 342, 99 and 27 males caught for Latin square replicate 1, 2 and 3, respectively. In total, MASTs set without a base captured 12 male *Ae. aegypti*, whereas MASTs set with bases of 40 cm, 60 cm and 80 cm caught 105, 157 and 199 males, respectively. Significant differences (ꭓ^2^ = 139.6, *df* = 3, *p* ≤ 0.05, *n* = 12) were noted between proportions of male *Ae. aegypti* caught in the MASTs with the four different base heights (Figure 2A). The MAST without a base caught significantly less males (0.58 ± 0.26 [mean ± S.E.]) than the MASTs set on a base which was 40 cm (8.75 ± 2.6), 60 cm (13.1 ± 4.8) and 80 cm (16.6 ± 7) high. Additionally, the proportions of males caught in the MAST set on a base which was 80 cm high were significantly greater than those in a MAST with a 40 cm or 60 cm high base. Trap location within the flight cage did not significantly influence male *Ae. aegypti* catch rates (ꭓ^2^ = 3.3, *df* = 3, *p* = 0.35, *n* = 12).

### 3.2. Experiment 2. MAST Base Height—Single Premises Field Trial

The MAST with the short base caught 86 male *Ae. aegypti* in total whereas the MAST with the tall base captured 74 males in total. No significant differences (ꭓ^2^ = 0.3, *df* = 1, *p* = 0.56, *n* = 12) were noted between two-hourly mean catches of male *Ae. aegypti* caught in the MAST with the short base (7.2 ± 1.4) and the MAST with the tall base (6.2 ± 1.2; Figure 2B). While trap location did significantly influence male *Ae. aegypti* catch rates (ꭓ^2^ = 15.8, *df* = 1, *p* ≤ 0.05, *n* = 12) time at which traps were run did not (ꭓ^2^ = 0.4, *df* = 1, *p* = 0.53, *n* = 12).

### 3.3. Experiment 3. MAST Base Height—Multiple Premisses

The MAST with the short base caught 38 male *Ae. aegypti* in total whereas the MAST with the tall base captured 123 males in total. This difference was largely attributed to one incidence where the MAST with the tall base caught 89 male *Ae. aegypti*. No significant differences (ꭓ^2^ = 0.18, *df* = 1, *p* = 0.67, *n* = 12) were noted between weekly mean catches of male *Ae. aegypti* caught in the MAST with the short base (3.2 ± 1.3) and the MAST with the tall base (10.25 ± 7.2; Figure 2C). Additionally, neither square (ꭓ^2^ = 3.8, *df* = 2, *p* = 0.15, *n* = 12) nor week (ꭓ^2^ = 5.2, *df* = 3, *p* = 0.16, *n* = 12) significantly influenced male *Ae. aegypti* catch rates during these trials. If the catch of 89 males is considered an outlier, removed and the data reanalysed, there is still no significant differences (ꭓ^2^ = 0.12, *df* = 1, *p* = 0.72, *n* = 11–12) between weekly mean catches of male *Ae. aegypti* caught in the MAST with the short base (3.2 ± 1.3) and the MAST with the tall base (3.09 ± 1.2; Figure 2D).

### 3.4. Experiment 4. MAST Head—Various Entrance Numbers Sizes and Shapes

In total 399 males were captured during these trials with 257, 96, and 46 males caught for Latin square replicate 1, 2, and 3, respectively. The MAST with the two 5 cm square entrances captured 101 males with a mean (±S.E.) catch rate per trial of 8.4 (±3.2; Figure 3A). The MAST with two 2.5-cm square entrances captured 86 males with a mean catch rate per trial of 7.2 (±2.1). The MAST with a single 2.5 cm square entrance captured 113 males with a mean catch rate per trial of 9.4 (±2.4). Finally, the MAST with the single 2.5-cm triangular entrance captured 99 males with a mean catch rate per trial of 8.3 (±2.0). No significant differences (ꭓ^2^ = 4.5, *df* = 3, *p* = 0.21, *n* = 12) were noted between mean proportional catches of male *Ae. aegypti* in the MAST versions (Figure 3A). However, trap location within flight cage did significantly influence male *Ae. aegypti* catch rates (ꭓ^2^ = 13.2, *df* = 3, *p* ≤ 0.05, *n* = 12) during these trials.

### 3.5. Experiment 5. MAST Head—Various Entrance Sizes

In total, 513 males were captured during these trials with 303, 133, and 77 males caught for Latin square replicate 1, 2, and 3, respectively. The MAST with an entrance with 1 cm sides captured 23 males in total, with a mean (±S.E.) catch rate per trial of 1.9 (±0.7). The MAST with an entrance with 1.5 cm sides captured a total of 51 males with a mean catch rate per trial of 4.3 (±1). The MAST with an entrance with 2 cm sides captured in total 147 males with a mean catch rate per trial of 12.3 (±2.8). Finally the MAST with an entrance with 2.5 cm sides captured 292 males with a mean catch rate per trial of 24.3 (±5.0). Male *Ae. aegypti* catch rates significantly varied between trap types (ꭓ^2^ = 115.7, *df* = 3, *p* ≤ 0.05, *n* = 12) with each larger entrance type catching significantly more male *Ae. aegypti* than the smaller entrance types (Figure 3B).

A consequence of reducing the entrance size was that the volume of sound emitted from the trap also decreased. Therefore, sound volume was also incorporated in this model and despite increases in mean volumes associated with larger entrances (55.3 dB (±0.4) for 1 cm entrances, 59 dB (±0.2) for 1.5 cm entrances, 60.9 dB (±0.4) for 2 cm entrances and 61.9 dB (±0.3) for 2.5 cm entrances) there was no significant variation in male catch rates due to this factor (ꭓ^2^ = 2.5, *df* = 1, *p* = 0.11, *n* = 12). Lastly, trap location within the flight cage also did not significantly influence male *Ae. aegypti* catch rates (ꭓ^2^ = 1.3, *df* = 3, *p* = 0.72, *n* = 12).

### 3.6. Experiment 6. Sound Lure—Continuous vs. Intermittent Tones

In total, 567 males were captured during these trials with 413 and 154 males caught for Latin square replicate 1 and 2, respectively. The MASTs with sound lures playing a continuous tone captured 305 males with a mean (±S.E.) catch rate per trial of 19 (±4.0). The MASTs with sound lures which played intermittent tones captured 262 males in total with a mean catch rate per trial of 16.4 (±3.1). Male *Ae. aegypti* catch rates did not significantly vary between trap types (ꭓ^2^ = 0.5, *df* = 1, *p* = 0.48, *n* = 16; Figure 4A). However, trap location within flight cage did significantly influence male *Ae. aegypti* catch rates (ꭓ^2^ = 19.6, *df* = 3, *p* ≤ 0.05, *n* = 16) during these trials.

### 3.7. Experiment 7. Sound Lure—Various Volumes

A total of 901 males were caught in this experiment with 676, 98, 40, and 87 caught for Latin square replicate 1, 2, 3, and 4, respectively. As numbers were dwindling in replicate 3 another spare ~200 males were released into the cage at the end of replicate 3.

The MASTs with sound lures playing an intermittent tone at 63 dB captured 291 male *Ae. aegypti* with a mean (±S.E.) catch rate per trial of 24.3 (±9.6). The MASTs with sound lures playing an intermittent tone at 68 dB captured 299 males with a mean catch rate per trial of 24.9 (±9.9). The MASTs with sound lures playing an intermittent tone at 74 dB captured 311 males with a mean catch rate per trial of 25.9 (±11.9). Male *Ae. aegypti* catch rates did not significantly vary between trap types (ꭓ^2^ = 1.17, *df* = 2, *p* = 0.56, *n* = 12; Figure 4B). However, trap location within flight cage did significantly influence male *Ae. aegypti* catch rates (ꭓ^2^ = 22.1, *df* = 2, *p* ≤ 0.05, *n* = 12) during these trials.

### 3.8. Experiment 8. Sound Lure—Various Frequencies

For male *Ae. aegypti* catches with all age groups combined, the MAST with a sound lure set to 450 Hz captured a mean (±S.E.) proportion of 0.79 (±0.02) and the MAST with a sound lure set to 500 Hz caught 0.82 (±0.02) males (Figure 4C). Subsequently, male *Ae. aegypti* catch rates did not significantly vary between trap types (ꭓ^2^ = 2.3, *df* = 1, *p* = 0.13, *n* = 23; Figure 4C).

Grouped by age, MASTs caught a mean (±S.E.) proportion of 0.70 (±0.03), 0.86 (±0.02), 0.81 (±0.02) and 0.83 (±0.03) male *Ae. aegypti* which were day 3, 4, 5, and 6 post-eclosion, respectively (Figure 4D). Subsequently, male *Ae. aegypti* catch rates were significantly influenced by mosquito age (ꭓ^2^ = 13.7, *df* = 3, *p* ≤ 0.05, *n* = 10–12; Figure 4D).

While replications were too few (*n* = 4) to warrant statistical analyses, 4 day post-eclosion male *Ae. aegypti* were captured at mean proportions of 0.90 (±0.04) and 0.85 (±0.01) when the sound lure frequencies were set to 550 and 600 Hz, respectively (Figure 4E). Males were not captured in the MAST when the sound lure was turned off (*n* = 4).

### 3.9. Experiment 9. Influence of Wind on Catch Rates

Wind treatments significantly influenced male *Ae. aegypti* catch rates in the MAST (ꭓ^2^ = 269, *df* = 3, *p* ≤ 0.05, *n* = 7–8; Figure 5). When wind wasn’t blowing on the MAST a mean (±S.E.) proportion of 0.85 (±0.04) male *Ae. aegypti* were captured, significantly more than any other treatment. When wind blew on the front of the MAST head continuously only 0.01 (±0.01) male *Ae. aegypti* were caught, significantly less than any other treatment. Lastly, no significant differences were found between male *Ae. aegypti* catch rates in the MAST when the wind either continuously blew on the back of the trap (0.44 ± 0.05) or intermittently blew on the front of the trap (0.32 ± 0.04).

## 4. Discussion

The prototype MAST developed is believed to be fit for purpose, as defined by the TPP. The MAST caught male *Ae. aegypti* using minimal power requirements and a passive trapping system. As the entrance is small and to the side, it offers protection for internal electrical components such as the sound lure and potential sensor and communication equipment. The MAST design achieved adequate standards of user acceptability regarding ease of use and annoyance from noise and odour emission as it is light, can stack and the sound lure attracts males at low decibels. Additional targets described within the TPP have been achieved in the field setting [17], where the trap was demonstrated to catch comparable numbers of male *Ae. aegypti* to the BG-Sentinel trap, without the vast majority of bycatch collection in the BG-Sentinel, and with specimens being retained in suitable condition for identification purposes after one week of deployment. Additionally, a version of the MAST was developed which did not require insecticides to be used during deployment [17]. Implications from key findings from trap optimisation trials are discussed below.

### 4.1. Base Height

The presence of a base and differences in base heights significantly influenced male *Ae. aegypti* catch rates in the MAST. Within the semi-field flight cage a very low proportion of males were caught in traps without bases, suggesting that the black base is a vital swarm marker to bring mosquitoes close to the traps even in confined settings with high mosquito abundances. Traps set on 80 cm high bases caught more males than those with shorter bases. However, catch rates of male *Ae. aegypti* in MASTs were not significantly influenced by variations between the shorter bases tested either in the semi-field cage or in the field. As male *Ae. aegypti* are attracted to dark objects of low reflectance [8,9,10], it stands to reason that a larger base may provide a stronger visual cue, especially from a distance. Additionally, as sound lures emit sound waves which do not travel very far [38], unless played at impractically loud [39] and repulsive [40] volumes, our results reinforce the importance of a dark visual cue to draw in males from a distance. These findings are consistent with Balestrino, et al. [41] who found that a dark swarm marker was also essential to attract male *Ae. albopictus* to the trapping area. However, the decision of what size base to deploy with the MAST may end up being an operational one as while larger bases may be more effective swarm markers an 80 cm base is potentially too large and cumbersome to be user-friendly, and would entail greater shipping costs.

### 4.2. Entrance Types

MASTs with single entrances displayed higher catch rates than MASTs with multiple entrances, but not significantly so. For these prototypes, it was likely that the shapes of the entrance did not greatly influence male behaviour and either a similar number of males entered the traps with larger entrances or more entered but also left within the short 5-min trial periods. A limitation to this study was that, as only single lures were ever used, for traps with two entrances the lures were facing between and not directly out of either entrance. Our observations suggest that when the male flight path crosses the direct ‘line of sound’ from the speaker they respond strongly to the sound lure and fly directly towards it. Therefore, directing the speaker towards the entrance is likely to be an important factor in increasing catch rates.

For MAST traps with triangular entrances with sides between 1–2.5 cm, male *Ae. aegypti* catches significantly reduced as entrance size decreased. Mosquito traps are generally designed with much larger entrances (the GAT’s entrance is 11.5 cm in diameter [42], the AGO has an entrance diameter of 12.8 cm [43], the BG-Sentinel trap entrance diameter is 11 cm). Our impressions from visual observations were that these mosquitoes hesitated to fly into small spaces, despite still catching some males with an entrance with only 1-cm sides. While sound volumes at trap entrances were associated with smaller entrances, there wasn’t a significant influence of this factor directly on male catch rates. Again, our personal observations noted a sudden change in flight path when the males appeared to fly into the trajectory of the sound emanating from the speaker. Therefore, the greater interference of the sound waves caused by smaller entrances may also have reduced the likelihood of the males’ flight paths crossing these attractive signals.

### 4.3. Sound Lure Settings

MAST can be deployed with sound lures that play tones intermittently (30 s on and off), rather than continuously, without a significant influence on male *Ae. aegypti* capture rates. This finding is consistent with the operational decisions by previous researchers using sound traps who suggested that male *Ae. aegypti* were adequately attracted to intermittent flight tones which significantly reduced battery consumption [26,27,44].

Our results suggest that MAST catch rates of male *Ae. aegypti* do not vary greatly as the sound lure volume changes between 63 to 74 dB at trap entrance. These findings are also consistent with previous deployments of sound traps with lure set between 60–70 dB used in a series of trials with the Sound-GAT by Johnson and Ritchie [25] and at 60 dB in additional Sound-GAT field trials by Staunton, et al. [44]. Researchers have also suggested that *Aedes* mosquitoes may even listen for their human hosts which normally speak at a sound level of 60–70 dB [39,45]. Consequently, these results support the operational decision to run sound lures at the lower volume settings in order to conserve power and be less likely to be heard by, and consequently irritate, residents.

Similarly, high catch rates of male *Ae. aegypti* occurred in the tent trials regardless of whether the sound lures frequencies were 450 Hz or 500 Hz at 29 °C. Additionally, although only a few trials were performed, males were still captured at high rates in MASTs with the sound lures set to even higher frequencies of 550 and 600 Hz and no males were caught when the sound lure was disabled. Previous laboratory trials performed by Costello [46] suggested that male *Ae. aegypti* were most responsive to tones played at 500 Hz for ambient temperatures of 27 °C or 31 °C. Later laboratory work by Brogdon [47] displayed a peak in female *Ae. aegypti* wing beat frequencies at 458–461 Hz when tested at 25 °C and semi-field trials performed by Johnson and Ritchie [25] found higher, but not significantly so, rates of males captured in Sound-GATs set to 484 Hz compared to 560 Hz. While our work is not inconsistent with previous trials, male *Ae. aegypti* did not demonstrate clear preferences for the frequencies tested.

A variety of studies have demonstrated that *Ae. aegypti* wing beat frequencies can change according to factors such as mosquito age, diet and subsequently size as well as the ambient temperature [25,27,48,49]. Our results suggest that the younger (3 days post eclosion) males are less responsive to MASTs with lures set to 500 Hz than older males. These findings potentially reflect previous work demonstrating that female *Ae. aegypti* wing beat frequencies increase during the first four days post-eclosion [40,46,49,50] and the further suggestion by Tischner and Schief [49] that males better respond to higher frequencies as they age during this period. These findings imply that young males may be less responsive to sound lures set in MASTs, relative to older cohorts. Depending on how long males survive in the field, the finding that younger males are less responsive to female wingbeat frequencies than older males may have important implications for rear and release programs releasing young males that are potentially less competitive or responsive to sound lures than older wild males.

### 4.4. Wind Effects

Lastly, wind negatively influenced catch rates of male *Ae. aegypti* in the MAST. The wind produced during these experiments measured 1.5 ms^−1^ which is the maximum flight speed previously recorded for *Ae. aegypti* [51]. Capture rates were also greatly dependent on the direction of the MAST entrance in relation to the wind. Practically no mosquitoes were caught when the MAST entrance faced the wind and collection rates were partly recovered when the MAST entrance faced away from the wind. This suggests that the trap itself acts as a wind break for male swarming which was consistent with our visual observations as well as reports by Amos, et al. [52] of greater male *Ae. aegypti* flight activity occurring on the down-wind side of the BGS Trap. Lastly, intermittent wind (1.5 ms^−1^ every 9 s) reduced the catch rate in the MAST by approximately 60% relative to those displayed during windless conditions. The finding that ~40% of the males normally caught without wind were caught in the 9-s intervals between intolerable winds demonstrates the strong drive that males have to swarm and respond to sound lures. Thus, during field use the MAST head entrance should be placed on the downwind side of the base, particularly if wind direction is consistent at a site, or in a location protected from strong winds.

## 5. Conclusions

The development of the Male *Aedes* Sound Trap was a complex process requiring the optimisation of multiple trap aspects. We first established a basic structural concept of the MAST which caught males and generally fit requirements outlined by the TPP, including user acceptability. We then optimised each component of the trap by observing and assessing the behaviour of males when parameters were adjusted. Observations of male swarming and their response behaviours were absolutely critical in indicating whether an adjustment was likely to improve or impede trap capture rates. Once promising prototypes were developed from comprehensive laboratory experimentation further assessments of trap performance occurred during extensive field trials in a variety of international locations involving male *Ae. aegypti* and *Ae. albopictus*, as well as non-target species [17,28,30]. The behavioural findings and developmental trap process described above are translatable to the development of other mosquito traps, especially that of *Aedes* sound traps, and can hopefully positively contribute to the improved surveillance of these vectors of disease.

## Figures and Tables

**Figure 1 insects-12-00388-f001:**
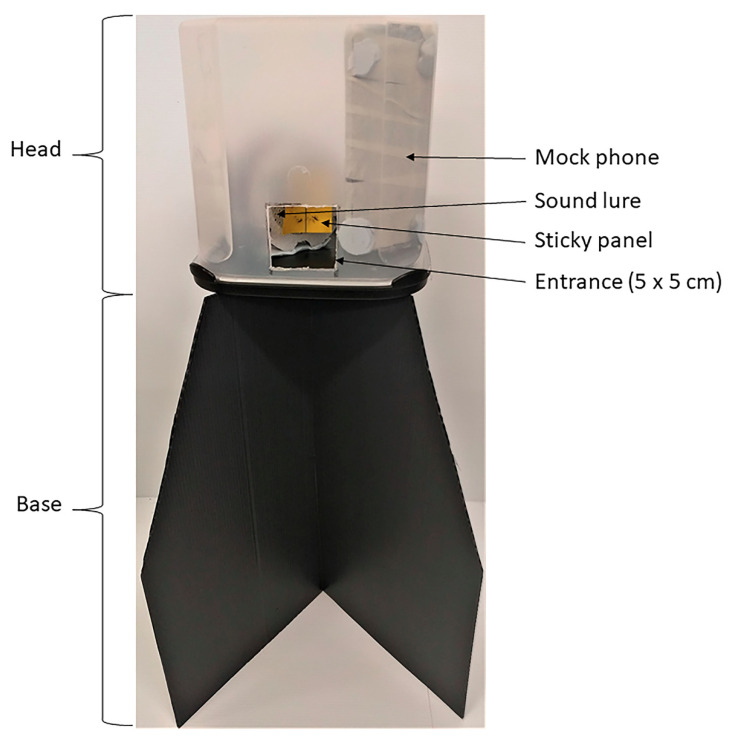
Basic general design of the main prototype MAST (MAST 1) used in developmental trials.

**Figure 2 insects-12-00388-f002:**
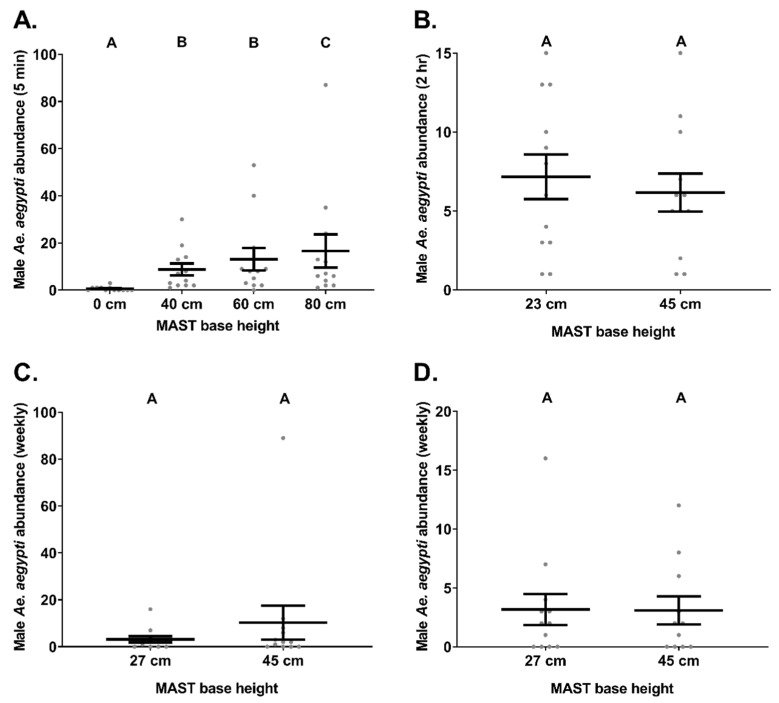
The influence of different base heights on male *Ae. aegypti* (mean ± S.E.) catch rates in the MAST. Trials investigated catches of male *Ae. aegypti* caught either (**A**) within the semi-field flight cage, (**B**) at a single field premises, (**C**) within multiple field premises (including the outlier) or (**D**) within multiple field premises (excluding the outlier). Different letters above points indicate significantly different groups (Tukey HSD, *p* ≤ 0.05, *n* = 12).

**Figure 3 insects-12-00388-f003:**
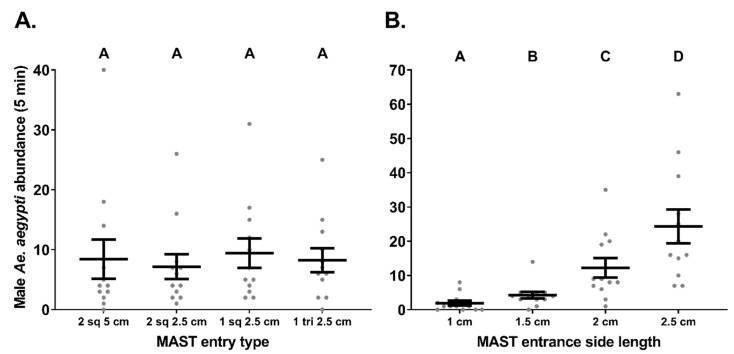
The influence of different entrance types on male *Ae. aegypti* (mean ± S.E.) catch rates in the MAST. Trials investigated catches of male *Ae. aegypti* (per 5 min) in semi-field trials investigating influences from different entrance (**A**) types and (**B**) sizes. Square entrances are referred to by “sq” and the triangular entrance is referred to by “tri”. The length of the sides of the entrance type is also noted in treatment labels. Different letters indicate significantly different groups (Tukey HSD, *p* ≤ 0.05, *n* = 12).

**Figure 4 insects-12-00388-f004:**
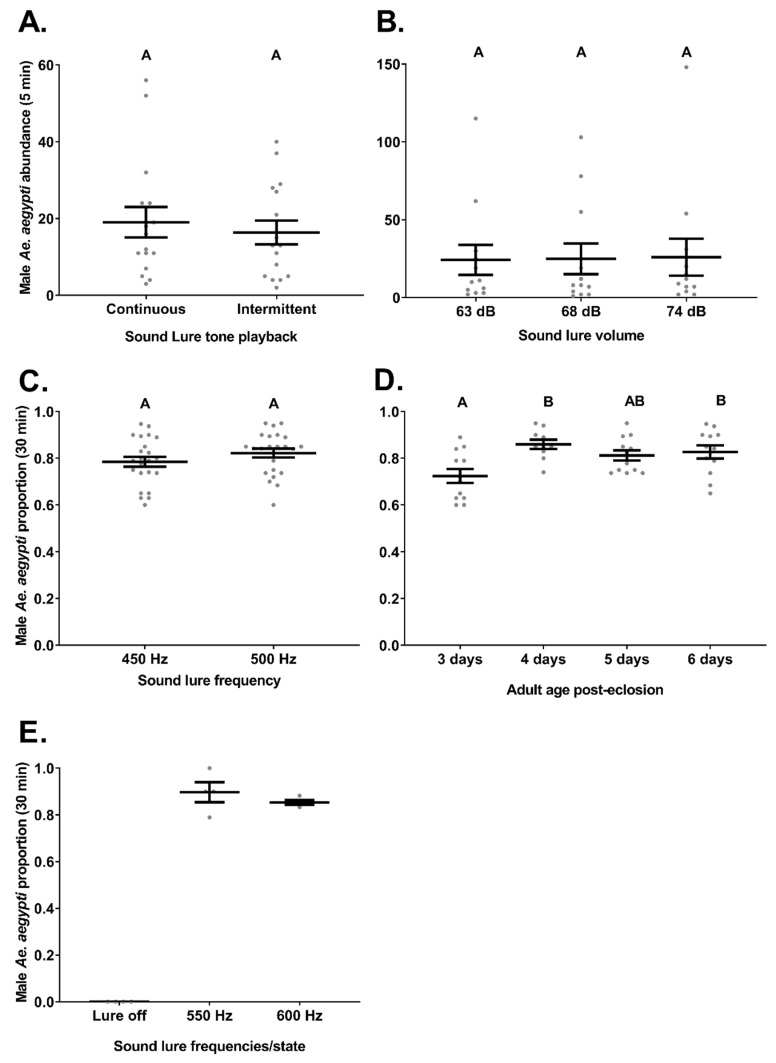
The influence of different sound lure configurations on male *Ae. aegypti* (mean ± S.E.) catch rates in the MAST. Trials investigated catches of male *Ae. aegypti* per (**A**) five-minute semi-field trials comparing male catches in MASTs (*n* = 12) with sound lures that either played continuously or intermittently (30 s on and off), (**B**) five-minute semi-field trials comparing male catches in MASTs (*n* = 12) with sound lures set to different volumes, (**C**) thirty-minute tent trials comparing male catches in MASTs (*n* = 23) with sound lures set to different frequencies, (**D**) thirty-minute tent trials comparing catch rates of males of various post-eclosion ages in MASTs (*n* = 10–12) set to 500 Hz and 60 dB, and (**E**) thirty-minute tent trials comparing male catches in MASTs (*n* = 4) with sound lures set to different frequencies or turned off. Different letters indicate significantly different groups (Tukey HSD, *p* ≤ 0.05).

**Figure 5 insects-12-00388-f005:**
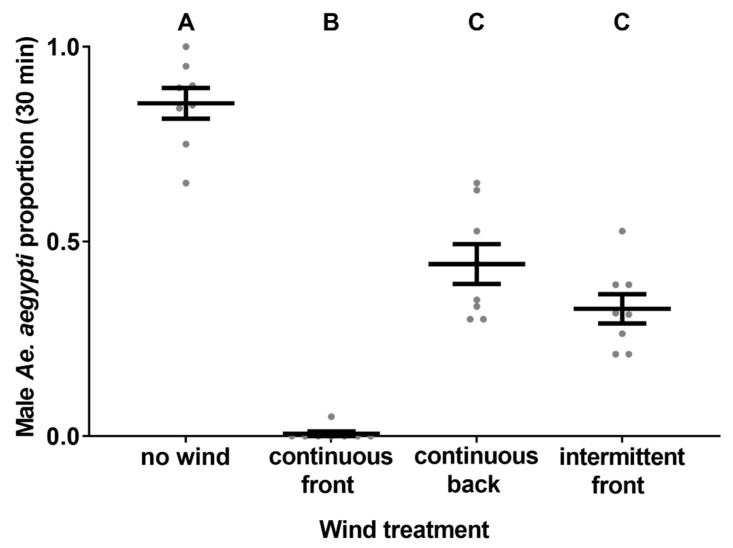
The influence of different wind conditions on male *Ae. aegypti* (mean ± S.E.) catch rates in the MAST. Different letters indicate significantly different groups (Tukey HSD, *p* ≤ 0.05, *n* = 7–8).

**Table 1 insects-12-00388-t001:** Target product profile for the development and validation of the male *Aedes* sound trap.

**Title**	Development and validation of a male *Aedes* sound trap	
**Category**	Vector surveillance	
**Disease**	Dengue, Zika, chikungunya, yellow fever	
**Grantee Organization**	Verily Life Sciences	
**Product Class**	New vector surveillance tool	
**Product Description**	A sound trap targeting male adult *Aedes aegypti* mosquitoes	
**Item**	**Desired Target**	**Minimally Acceptable Standard**
**Indication**	Develop novel entomological surveillance tool to trap male *Aedes* mosquitoes (especially *Ae. aegypti*) comparable to the catch rates in adult traps in use (e.g., Biogents Sentinel trap 2)	Develop novel entomological surveillance tool to trap male *Aedes* mosquitoes, especially *Ae. aegypti.*
**Target Human Population (s)**	All age groups and populations in vector borne disease endemic countries in key space	All age groups and populations in vector borne disease endemic countries in key space
**Application Method**	For use in a variety of surveillance programs	For use in a male release programs throughout urban landscapes
**Safety**	Trapping device safe for use in or near human dwellings	Is not an electrical or fire hazard, does not include the use of non-WHO recommended insecticides
**Expected Performance:**	Male *Ae. aegypti* catch rates exceed other adult traps in use (e.g., Biogents Sentinel trap 2, Sound BG-GAT)	Male *Ae. aegypti* catch rates are comparable to other adult traps in use (e.g., Biogents Sentinel trap 2, Sound-GAT)
	Trap uses small (e.g., AA) batteries which last long periods of time (e.g., 1–2 months)	Trap uses small (e.g., AA) batteries which last one week
	Specimens are retained in suitable condition for morphological identification and/or genetic analysis after 1–2 months of deployment	Specimens are retained in suitable condition for morphological identification and/or genetic analysis after 1 week of deployment
	Trap design is able to support sensors and protect them from environmental conditions even in unsheltered locations	Trap design is able to support sensors, but must be placed in sheltered location to protect sensors from environmental conditions
	Trap design supports varying audio levels, frequencies (single and sweeps) and programmable for timed operations to enable targeting of different *Aedes* species	Trap design supports varying audio levels and frequencies to enable targeting of different *Aedes* species
**Nontarget organisms and environmental risk Assessment**	The proportion of nontarget organisms captured in this trap is <1% and are able to be identified by sensor equipment	The proportion of nontarget organisms captured in this trap is <5% and are able to be identified by sensor equipment
	The abundance of nontarget organisms killed in this trap is significantly less than other traps using fan capture systems	Acceptable risk to non-target organisms and environment when product is used according to directions
	Trap will not act as larval habitat	Trap able to be treated so that it remains as a non-productive larval habitat
**Freedom to operate**	Commercialisation controlled by Verily Life Sciences	Patented by Verily Life Sciences
**Shelf life/storage stability**	Product able to be stored for years without reduction in performance	Product able to be stored for duration comparable with other mosquito traps
**User acceptability**	Trap design facilitates reasonable ‘ease of use’ so that it is deployable and easily serviceable by a range of general surveillance staff	Trap design facilitates reasonable ‘ease of use’ so that it is deployable and serviceable so that is suitable to male release programs
	Very quiet, but effective noise (sound lure not irritable to occupants)	Acceptable noise (sound lure not irritable to occupants)
	No smell (odour from lures not irritable to occupants)	Acceptable smell (odour from lures not irritable to occupants)
**Target price**	Significantly less expensive than other mosquito traps, especially when deployed at scale	Commercially competitive with other mosquito traps, especially when deployed at scale
**Challenges/Risks**	Trap components unattractive to community and unlikely to be tampered with.	Trap components are still attractive to community and therefore may be tampered with, but are easily replaceable.
	Effective on limited spectrum of *Aedes* spp.	Effective collection method on key regional *Aedes* spp.

## Data Availability

The datasets used and/or analysed during the current study are available from bradwhite@verily.com at Verily Life Sciences on reasonable request.

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
