# Peer review of "Designing Aedes (Diptera: Culicidae) Mosquito Traps: The Evolution of the Male Aedes Sound Trap by Iterative Evaluation"

_insects, 2021, doi:10.3390/insects12050388_

Round 1
Reviewer 1 Report
Line 85, "Fans are extremely effect..." should be "Fans are extremely effective..."
I don't consider the BG-2 trap to be a sound trap.
Why did you not include Pantoja-Sanchez et al 2019, Journal of Vector Ecology 44(2): 216-222 paper in your citations?
I what the relative importance of visual and sound are?
Reviewer 2 Report
Dear authors
the article reports the research activities for the development of a sound trap already published by the same authors in 2020. The article is therefore not extremely original but shows the research path undertaken by the authors to evaluate the individual components of the MAST trap. In my opinion, the article should include some notes in the introduction to emphasize how both females and males can be attracted to sound frequencies and it is also necessary to underline that Aedes males have been observed to respond to a wide sound range such as to exclude the possibility of creating extremely selective species-specific sound traps.
In the Methods the collection of mosquitoes was described as a knock down method with CO2. However, in the standard trap design the collection system is not described and in Figure 1 is reported the presence of a sticky panel. It is necessary to clarify in which way the mosquitoes were collected and trapped. In order to understand the article, it is also necessary to insert in the trap design a short description of the sound system used in the traps even if it is already detailed in the article of 2020. It is also necessary to integrate in the main text the sound emissions details (only reported in table S_1: Sound lure settings) in the description of the different experiments or in the paragraph 2.1.
The description of the experiments often reports indications on the statistical analysis carried out which should instead be moved to paragraph 2.3 (Standard data analysis). It may also be useful to move this paragraph to the end of the Methods chapter after the description of the experiments.
In the discussion I have included notes and text taken from the final discussion of the article Balestrino et al. 2016. Acta Trop. 164: 448–454 because it reports similar results and mentioned many references of interest not cited by the authors and which should in my opinion be included or considered. The similar results obtained by all these authors should be emphasized or compared particularly with regards to the importance of sound volume, the need to integrate black visual stimuli, the need to continue to work not only on sound frequencies but also on the emission mode and on the different sound response of males according to age.
The different responsiveness of males at different age raises an important issue to be discussed / cited especially with regard to the need to introduce SIT/IIT males of adequate age to be effectively competitive with the wild counterparts.
These and other small notes are reported in detail in the attached pdf file.

Reviewer 3 Report
Dear Authors
Aedes aegypti is the primary vector of a number of viruses such as Zika, dengue or yellow fever and its current distribution is of major concern. Therefore, effective surveillance is of outmost importance.
I consider that this study is well designed and the results are congruent. Traps targeting male mosquitoes can be a good alternative to detect species presence. Moreover, with the development of new mass rearing and release control programs, there is a special interest in assessing male densities in the field.
The manuscript is well written and reflects planning and labor. As the authors I also believe this work is “applicable to the development of alternate mosquito traps beyond Aedes sound traps and provides useful information towards the improved surveillance of these disease vectors.”
I would recommend a revision of the text in order to fix some typos.
E.g.: in different places of the manuscript, including table S1, sometimes we can see the word “premisses” and other times “premises”. Please revise.
E.g.: page 7, line 216: where we read “were rotate”, I believe we must read “were rotated”. This happens in other sentences.
